# A Frame-by-Frame Glance at Membrane Fusion Mechanisms: From Viral Infections to Fertilization

**DOI:** 10.3390/biom13071130

**Published:** 2023-07-14

**Authors:** Farshad C. Azimi, Trevor T. Dean, Karine Minari, Luis G. M. Basso, Tyler D. R. Vance, Vitor Hugo B. Serrão

**Affiliations:** 1Department of Biochemistry, University of Toronto, Toronto, ON M5S 1A8, Canada; farshad.azimi@utoronto.ca; 2Pharmaceutical Sciences, University of Illinois Chicago, Chicago, IL 60612, USA; tdean8@uic.edu; 3Biomolecular Cryo-Electron Microscopy Facility, University of California-Santa Cruz, Santa Cruz, CA 95064, USA; karineminari@gmail.com; 4Laboratório de Ciências Físicas, Universidade Estadual do Norte Fluminense Darcy Ribeiro, Campos dos Goytacazes, Rio de Janeiro 28013-602, Brazil; luisbasso@uenf.br; 5Department of Laboratory Medicine and Pathobiology, Temerty Faculty of Medicine, University of Toronto, Toronto, ON M5S 1A8, Canada; tyler.vance@utoronto.ca; 6Department of Chemistry and Biochemistry, University of California-Santa Cruz, Santa Cruz, CA 95064, USA

**Keywords:** membrane fusion, fertilization, viruses, cryo-electron microscopy, structural biology, fusion mechanism

## Abstract

Viral entry and fertilization are distinct biological processes that share a common mechanism: membrane fusion. In viral entry, enveloped viruses attach to the host cell membrane, triggering a series of conformational changes in the viral fusion proteins. This results in the exposure of a hydrophobic fusion peptide, which inserts into the host membrane and brings the viral and host membranes into close proximity. Subsequent structural rearrangements in opposing membranes lead to their fusion. Similarly, membrane fusion occurs when gametes merge during the fertilization process, though the exact mechanism remains unclear. Structural biology has played a pivotal role in elucidating the molecular mechanisms underlying membrane fusion. High-resolution structures of the viral and fertilization fusion-related proteins have provided valuable insights into the conformational changes that occur during this process. Understanding these mechanisms at a molecular level is essential for the development of antiviral therapeutics and tools to influence fertility. In this review, we will highlight the biological importance of membrane fusion and how protein structures have helped visualize both common elements and subtle divergences in the mechanisms behind fusion; in addition, we will examine the new tools that recent advances in structural biology provide researchers interested in a frame-by-frame understanding of membrane fusion.

## 1. The Biophysical Process of Membrane Fusion

Membranes are essential structures that define the boundaries of cells and organelles and play crucial roles in many cellular processes, such as transport, signaling, and cell–cell interactions. The structure of membranes is complex and dynamic, consisting of a phospholipid bilayer with embedded proteins and other lipids. The properties of the membrane, such as fluidity, thickness, and curvature, are determined by the composition and organization of these components [1,2].

Membrane fusion is a complex process that involves the merging of two lipid bilayers to form a single continuous membrane structure. The following steps outline the biophysical process of membrane fusion (Figure 1):*Recognition and contact*: Membrane fusion requires two membranes to come into close proximity. This initial step is often facilitated via specific protein receptors on the surface of the membranes that recognize each other and bring the membranes into contact.*Hemifusion*: During this step, the outer leaflets of the two membranes begin to merge and form a stalk-like intermediate structure called a hemifusion diaphragm. This is a highly energetically unfavorable process, as it requires the hydrophobic tails of the lipids to come into contact with water.*Expansion*: The hemifusion diaphragm then expands to create a larger opening between the two membranes, allowing for the mixing of their contents.*Full fusion*: The final step involves the complete merging of the two membranes to form a single continuous bilayer. This process is also energetically unfavorable and requires the formation of a fusion pore, which allows for the exchange of membrane components.

Membrane fusion is governed by both biophysical and thermodynamic factors, which are intertwined and contribute to the overall process [1,2,3].

*Biophysical factors*: Biophysics is the field of study that explores the physical and chemical properties of biological molecules and systems, including their structures, dynamics, and interactions. Understanding the biophysics of membranes is crucial for unraveling their role in various cellular processes, including membrane fusion [3]. Membrane fusion is dependent on the interactions between the proteins and lipids on the surface of the target membranes, as well as the properties of the membranes themselves, such as their curvature, fluidity, and elasticity. The overall process of membrane fusion can be thought of as a series of energy-dependent steps that involve changes in membrane curvature, lipid mixing, and the formation of a fusion pore [4]. Understanding these factors is essential for developing a comprehensive understanding of the mechanism of membrane fusion [3,5,6,7,8,9,10].

*Thermodynamic factors*: The thermodynamics of membrane fusion is determined by the free energy changes associated with the process. The free energy changes can be analyzed using thermodynamic models, such as the free energy landscape, which considers the free energy contributions from various sources, including the protein–protein and protein–lipid interactions, the elastic deformation of the membranes, and the entropy changes associated with the mixing of lipids. One of the key factors that govern the thermodynamics of membrane fusion is membrane tension. Membrane tension is the force per unit length that is exerted on the membrane from the internal and external environment [4,5]. High membrane tension can result in unfavorable free energy changes and hinder membrane fusion, whereas low membrane tension can facilitate fusion.

Recent advances in imaging and computational techniques have enabled detailed studies of membrane structure and dynamics, as well as the mechanisms of membrane fusion. For example, molecular dynamic simulations have been used to simulate membrane behavior and the dynamics of membrane fusion [4]. Furthermore, structural biology has been used to elucidate snapshots during membrane fusion, and most recently, cryogenic electron microscopy (cryo-EM) has been extensively used to visualize the structures of membrane fusion proteins at high resolution, providing insights into their mechanisms of action [6,11,12,13,14,15,16,17,18,19,20,21,22,23,24].

## 2. Mechanisms of Membrane Fusion in Biological Systems

The thermodynamic hurdles that block membrane fusion are vital for biological life as we know it. However, many biological processes call for the fusion of membranes, requiring a regulated mechanism that satisfies the thermodynamic and kinetic requirements to bring membrane fusion onto a biological timescale. Examples of such processes can be found in reproductive biology with both the fusion of male and female gametes and the resulting creation of the placental syncytiotrophoblast layer, developmental biology in the formation/repair of muscle fibers, and the pathobiological contexts of the reproductive life cycle of parasites and the entry of enveloped viruses into host cells.

All of the above-mentioned biological mechanisms are geared towards membrane fusion and require performing two tasks. First, the mechanism must bring the opposing membranes into very close proximity, overcoming the repulsive forces that exist between the phospholipid head groups and other surface molecules. Second, the mechanism must distort and destabilize the bilayers, making a single membrane more energetically favorable than two. Organisms have evolved a variety of mechanisms to complete these dual tasks of membrane fusion, the majority requiring the presence of fusion proteins, also known as fusogens.

Fusion proteins are membrane-spanning proteins with extracellular domains that catalyze membrane fusion. Despite the conserved name, fusogens can come in many different shapes and sizes, leading to three general strategies of fusogen-mediated fusion:*Multi-Membrane Complex Fusogens*: Fusogens can be protein subunits on opposing membranes that come together to form a complex, thereby dragging the membranes into close proximity. The SNARE protein complexes that catalyze fusion between intracellular vesicles/organelles are an obvious example of such fusogens, as discussed in Section 3.*Small Membrane-Destabilizing Fusogens*: Fusogens can be small transmembrane proteins capable of destabilizing membranes but relying on a combination of adhesion proteins and cytoskeleton-mediated membrane protrusions to facilitate close proximity. Examples of such fusogens will be discussed in Section 4.*Large Structurally Dynamic Fusogens*: Finally, fusogens can be large, dynamic proteins that undergo conformational changes in order to extend outwards from one membrane, insert a hydrophobic peptide into an opposing membrane to both anchor to it, disrupt its fluidity, and then pull the two membranes into close proximity. These types of fusogens have been extensively researched in viral entry (Section 5), with structurally homologous proteins appearing in eukaryotic processes as well (Section 6).

The mechanistic details behind these processes have been readily studied using a combination of scientific techniques and disciplines. However, our knowledge of the intricate details behind these strategies is highly indebted to structural biology and the art of protein structure determination. In this review, we will examine fusogen-mediated membrane fusion and how structural biology has shaped, and will continue to shape, our understanding of it.

## 3. Intracellular Trafficking and Complex-Forming Fusogens: SNARE Proteins

The intracellular trafficking of biological membranes refers to the movement of membrane-bound organelles and vesicles within a cell, which is critical for protein and lipid transport, secretion, and endocytosis. This trafficking process is essential for maintaining the proper distribution and function of cellular components, and dysregulation of intracellular trafficking is associated with various diseases [25,26,27,28].

The trafficking of biological membranes is a complex process that can be broken down into three steps: vesicle formation, vesicle transport, and vesicle fusion. Vesicle formation is mediated via the recruitment of cargo molecules into the membrane, which is facilitated by various coat proteins such as clathrin and Coat Protein Complex II (COPII). These coat proteins help shape the membrane into a vesicle by forming a cage-like structure around the cargo molecules [29,30,31,32,33]. Vesicle transport is mediated via motor proteins, such as kinesin and dynein, which move the vesicles along microtubules or actin filaments. These motor proteins bind to receptors on the vesicle surface and generate the necessary force to move the vesicles along the cytoskeleton [27,34,35,36,37]. Finally, vesicle fusion is the process by which the vesicle membrane fuses with the target membrane, allowing the cargo molecules to be released into the target organelle or onto its surface. This process is mediated via the formation of a fusion complex between the vesicle and target membranes, which is facilitated by various proteins, such as SNAREs [38,39,40,41,42,43,44,45,46]. These proteins help to overcome the energetic barriers involved in membrane fusion and ensure that the process is tightly regulated (Figure 2).

SNAREs, also known as Soluble NSF Attachment Protein Receptors, are a family of proteins that play a critical role in the fusion of vesicles with their target membranes in intracellular trafficking. From a structural perspective, SNAREs have a highly conserved coiled-coil structure that allows them to interact and form a stable complex, also known as the SNARE complex [39]. The SNARE complex is formed via the interaction of v-SNAREs and t-SNAREs, which are present on the vesicle and target membranes, respectively. The fusogenic domains of v-SNAREs consist of a single α-helix, while the coiled-coil domain of t-SNAREs consists of two α-helices in a complex with a third α-helix from Syntaxins, all three of which wrap around the v-SNARE helix. The interaction between the v-SNARE, Syntaxins, and t-SNARE domains forms a four-helix bundle, which is thought to be the driving force behind membrane fusion [39]. The SNARE complex has a conserved architecture known as the SNAREpin, which is composed of four α-helices that form a parallel bundle [40,41].

The SNARE complex is highly stable and resistant to disassembly, which is essential for efficient vesicle fusion. The disassembly of the SNARE complex is mediated by a group of proteins known as NSF (N-ethylmaleimide-sensitive factor) and SNAPs (SNARE-associated proteins), which use ATP hydrolysis to dissociate the SNARE complex and recycle the SNARE proteins for subsequent rounds of vesicle fusion [38,39,40,41,42,43].

The regulation of intracellular trafficking is critical for maintaining the proper distribution and function of cellular components. The dysregulation of this process can lead to various neurodegenerative disorders and cancer. For example, defects in the trafficking of synaptic vesicles have been linked to Alzheimer’s and Parkinson’s disease [47,48]. New developments in structural biology, such as improvements in cryo-EM, have been revolutionizing membrane fusion visualization, allowing the in situ or high-resolution determination of SNARE complexes [39,49].

## 4. Small Membrane-Destabilizing Fusogens

Going from vesicular fusion to fusion between cells, FAST proteins are small single-pass transmembrane proteins produced by a variety of reoviruses that are able to induce fusion between host cells, thereby giving their viral benefactor access to additional cells via direct cell-to-cell transmission [50]. Six types of FAST protein have been discovered to date, each made up of three general regions (ectodomain, transmembrane, and endodomain). All three domains are vital to fusion and partially interchangeable, allowing for swapping between the FAST proteins without loss of function [50].

FAST proteins form multi-protein complexes at distinct zones on the membrane; it is thought that these complexes modulate membrane fluidity by producing distorting dimples [51]. However, the ectodomains of FAST proteins are small, begging the question of how these proteins would be able to even reach an opposing membrane. Interestingly, the p14 FAST protein has been seen to stimulate actin polymerization with a phosphorylated tyrosine in its endodomain [52]. The actin produces a membrane protrusion that pushes the small ectodomain into close contact with an opposing membrane. From there, host adhesion proteins are required to maintain close proximity [53].

The use of the cytoskeleton to produce fusion-seeking protrusions is not unique to p14. In *Drosophila* muscle syncytia formation, the two fusing muscle cells are structurally distinct from one another, with one forming an invading protrusion and the other forming a cytoskeletal mesh that braces against the protrusion [54]. This has led to a three-step mechanism, where the two cells adhere to each other, produce the aforementioned cytoskeleton-based invasion/resistance, and then promote lipid bilayer destabilization by unknown means [55]. While myoblast fusion in vertebrates is known to be associated with cytoskeleton restructuring, it is unclear how universal the *Drosophila* strategy is across the Tree of Life. That said, the proteinaceous component responsible for membrane destabilization in vertebrates is known, namely myomaker.

Myomaker is a multi-pass protein on the surface of myocytes that is vital for fusion in both muscle development [56] and repair [57] in several model organisms, including both mice and zebrafish [56,58]. Similar to FAST proteins, myomaker has a relatively little reach away from the cell surface, with its ectodomain composed of 3- to 14-amino acid loops that connect the transmembrane helices [59]. AlphaFold prediction shows these loops to be mostly unstructured, save for a small potential helix (AFDB: A6NI61). Myomaker would likely require cytoskeletal intervention in order to reach the opposing membrane.

Unlike FAST proteins, myomaker does not work alone. Fibroblasts expressing myomaker endogenously on the surface of fibroblasts are able to fuse with myocytes but not other myomaker-expressing fibroblasts [56]. Myocytes must, therefore, contain an additional component that facilitates myomaker’s function. This unknown factor was discovered to be the surface protein myomerger, also known as myomixer or minion [58,60,61]. The two proteins work together to promote fusion, with myomaker being responsible for the formation of the hemifusion intermediate, while myomerger is needed for the collapse of the hemifusion intermediate into a fusion pore [62].

## 5. Structurally Dynamic Fusogens—Viral Entry

### 5.1. What Is Viral Fusion?

Another important biological event that requires membrane fusion is enveloped virus entry/infection [63]. Enveloped viruses are enclosed in a phospholipid bilayer that originates from the cellular membranes of previously infected cells; the viral membranes form as the nascent virions bud off the host cell surface and are released into the extracellular environment. From there, the virions must find another host cell and begin the process anew via viral entry, a membrane fusion event that proceeds in the opposite direction of the viral release process. In this way, viral fusion and viral release are two faces of the same biological coin [63].

Viral release occurs in the late viral replication cycle when the viral genome and structural components (particularly the viral fusion proteins) have accumulated to critical concentrations in the cell’s cytosol and cellular membranes. This high concentration of viral structural elements drives the viral release process forward in a thermodynamically favorable fashion [64,65]. On the contrary, viral fusion occurs at low protein-concentration regimens, where viral particles exist in dilute concentrations in the extracellular environment. Therefore, the virus–cell interface (sometimes referred to as virus–cell synapse) relies on high-affinity interactions between the virus and the cell. These interactions are mediated by the viral fusion proteins (or sometimes a secondary viral membrane protein) and their binding protein counterparts on the host cell, commonly referred to as receptors (and in cases further supported by co-receptors). At a mechanistic level, the thermodynamic, kinetic, and structural determinants of the protein–protein interaction events between the virus and the cell inform all other aspects of the fusion process. Many of these aspects are shared among all enveloped viruses, while some are unique to specific viral classes.

### 5.2. Shared Aspects of the Viral Fusion Processes of Enveloped Viruses

Enveloped viruses cause some of the most common and impactful diseases of the modern era, including COVID-19, acquired immunodeficiency syndrome (AIDS), and seasonal influenza [63]. Parallel to their heterogeneity in cellular tropism, symptoms, and disease progression, enveloped viruses demonstrate vast heterogeneity in their virion structure, including their fusion proteins and subsequent processes. While this review article will tease apart the mechanistic details of the fusion process among specific classes of enveloped viruses, it is important to emphasize that all classes share a common viral fusion process [64]. There is a natural chronology to this process that can be divided into three stages based on the fusion state of the viral and host cell membranes: the pre-fusion stage, the intermediate events leading to membrane fusion, and the post-fusion stage (Figure 3).

*Pre-fusion State*: At the outset, the viral and host cell membranes are separate yet opposed within immediate proximity of one another, with the viral fusion proteins existing in pre-fusion homo-oligomeric complexes (either homodimers or homotrimers, depending on the virus). This proximity allows for an attachment event between a receptor protein on the target cell membrane and either the receptor-binding domain of the viral fusion protein or another membrane-integral protein of the virus. The protein–protein interaction keeps the two membranes anchored to one another for the following stages.

*Intermediate States*: The attachment event, either with or without an accompanying drop in pH, provides the required “trigger” for the homo-oligomeric pre-fusion protein complexes to undergo a drastic conformational rearrangement. This conformational rearrangement results in the partial insertion of the fusion subunit of the viral fusion protein into the opposing cell membrane. This region is called a fusion peptide or a fusion loop and possesses a significant hydrophobic character, as described in the next section. By the end of this step, the homo-oligomeric viral fusion protein complex exists in an extended state, bridging the two apposed membranes with the N-terminal fusion peptide inserted into the host membrane and the C-terminal transmembrane region (TM) of the fusion protein embedded in the viral membrane. This state is commonly referred to as the pre-hairpin state (as a later rearrangement result in hairpin topologies) and is almost always homo-trimeric in nature, regardless of the oligomeric state of the viral pre-fusion protein complex. In this state, the extended trimeric fusion complex has its N- and C-termini juxtaposed to the host and viral membranes, respectively.

*Post-fusion State*: In an enthalpically favored event, one or several pre-hairpin complexes undergo internal bundling, where each protomeric unit of the trimeric pre-hairpin fusion complex bends in the middle and collapses its N- and C-termini toward one another, forming an energetically stable trimer-of-hairpins, known as the post-fusion state, and pulling the viral and cellular bilayer membranes towards one another in the process. As the two membranes are pulled together, the outer leaflets fuse and form the hemifusion state of the membranes and then continue towards a full fusion state, where the two membranes become contiguous, resulting in the fusion pore. Finally, the fusion pore expands and widens, through which the viral genome and other contents are released into the intracellular space [64].

### 5.3. Specific Aspects of the Viral Fusion Process Observed among Three Viral Fusion Classes

It is through the fundamental fusion process described above that the inherent repulsive interactions between viral and host membranes are overcome, and two separate membranes fuse into one. However, the structural details of an enveloped virus’s fusion protein can impact the mechanistic specifics of this process, allowing for viral grouping into one of three classes: class I, class II, and class III [65,66].

*Class I:* Class I viral fusion occurs in many of the most clinically significant and well-known viral species (e.g., influenza viruses, HIV, and Ebola virus) and was, naturally, the first to be studied. Class I viral fusion is mediated by homo-trimeric pre-fusion protein complexes that need to undergo proteolytic activation before they are fusion competent. This proteolytic activation occurs either within the secretory pathway of a previously infected host cell (e.g., for HIV) [67] or extracellularly (e.g., for influenza A virus) [68]. Regardless of the location, proteolytic activation is indispensable to viral infectivity within this class. In fact, proteolytic activation can inform the virulence of an enveloped virus. For example, compared to the non-virulent influenza A viruses, the pathogenicity of influenza strains H5 and H7 is correlated with the ease of proteolytic activation of the fusion protein via systemically expressed furin-like proteases [65]. It should be noted that the two polypeptide fragments resulting from proteolytic processing events remain covalently associated via inter-polypeptide disulfide bonds.

In class I fusion processes, the event that triggers the conformational change in viral fusion proteins is acidic pH, direct receptor binding with or without acidic pH, or indirect receptor binding through a separate viral membrane protein called the attachment viral protein. The mechanism involving indirect receptor binding is noteworthy in that it requires the hetero-oligomerization of the attachment and fusion of viral proteins prior to the attachment event. However, after viral protein–cell interaction, the attachment and fusion proteins/complexes detach, allowing the fusion proteins to form their characteristic pre-fusion homotrimers.

Regardless of the nature of the triggering event, class I viral fusion proteins possess a hallmark structural arrangement of two heptad repeat (HR) regions: an N-terminal HR region located immediately downstream of the membrane-inserting fusion peptide (FP) and a C-terminal HR region located immediately upstream of the TM region. HR regions contain multiples of seven amino acid motifs possessing hydrophobic amino acids in fixed repetitive positions. It is the collapsing of these two sets of HR regions towards one another that results in the post-fusion coiled-coil conformations of class I viral fusion proteins. The structure resulting from this collapse is called a six-helix bundle (6HB), the formation of which pushes the host and viral membranes toward their inevitable fused fate.

*Class II:* Class II viral fusion occurs in flaviviruses, alphaviruses, and bunyaviruses [69,70], with the latter viral class being evolutionarily distinct from the other two [69]. A hallmark feature of this class is the diverse oligomeric state of the viral fusion protein in the pre-fusion state. For example, in alphaviruses, viral fusion protein forms a heterodimer with another viral protein that chaperones the fusion protomer through the secretory pathway and, in some cases, forms trimers of dimers [71], while in flaviviruses, the viral fusion proteins form a dimeric complex in the prefusion state [72]. On the other hand, class II pre-fusion viral fusion proteins become fusion-competent via a different mechanism than that of class I. The nascent class II viral fusion proteins form heterodimers with a chaperone viral protein within the endoplasmic reticulum. This chaperone requirement may potentially be due to the primarily β-sheet character of class II viral fusion proteins, as β-sheet proteins tend to fold slower [70], versus the primarily α-helical character of class I and the α/β character of class III fusion proteins. Within the secretory pathway, the acidic pH environment of the trans-Golgi apparatus will induce structural rearrangement of the class II viral fusion proteins into a homodimer, releasing the viral chaperone protein to be digested by resident furin-like proteases. In contrast to class I, mature class II viral fusion proteins exist in homo-dimeric complexes on the virion surface, remaining uncleaved prior to the attachment. This mature homo-dimeric state runs parallel to the viral membrane surface, a topology observed only in class II members.

Class II viral fusion proteins are triggered by acidic pH in the endosome. Initially, the hetero- or homo-dimeric complexes of the viral fusion protomers dissociate, followed by a rearrangement of the TM regions and fusion loops into a topology perpendicular to the viral membrane surface, similar to class I viral GPs. In this topology, the protomers form a homo-trimeric fusion-competent complex while inserting the fusion loops into the host cell membrane, resulting in the characteristic pre-hairpin conformation. Similar to class I members, the resolution of this pre-hairpin conformation to a hairpin conformation drives the two membranes toward the fused state. In the post-fusion state, the class II viral fusion protein complexes retain their primarily β-sheet character, in contrast to the α-helical character of the 6HB in class I [73]. As mentioned earlier, in alphaviruses, the mature fusion protein maintains a hetero-dimeric state with another viral protein that serves as the site of receptor attachment [65]. Acidic pH promotes the dissociation of this receptor attachment moiety from the fusion protein, allowing the viral fusion proteins to proceed with the general theme explained above.

**Table 1 biomolecules-13-01130-t001:** Viral fusion classes: a summary view.

	Class I	Class II	Class III
Pre-fusion State	Homo-trimeric ^a1^ (or Hetero-dimeric ^a2^)	Hetero- or Homo-oligomeric ^b1, b2^	Homo-trimeric
Proteolytic Processing of Fusion Proteins	Yes	No ^c^	No (or not needed for infectivity)
Metastability	Yes	Yes	No (reversible equilibrium)
Orientation to the Viral Envelope	Perpendicular	Parallel	Perpendicular
Secondary Structure Composition	α-helical	β-sheets	α-helical + β-sheets
Triggering Factors	Diverse ^d^	Acidic pH	Diverse ^e^
Post-fusion State	Homo-trimeric	Homo-trimeric	Homo-trimeric

Table adapted and modified from White et al. [64]. ^a1^ When the attachment moiety resides in the viral fusion protein. ^a2^ When the attachment moiety resides in a separate viral membrane protein. This attachment viral protein forms a heterodimer with the viral fusion protein. ^b1^ The viral fusion protein forms a heterodimer with another viral protein while traversing the cellular secretory pathways. These dimeric complexes have been shown to trimerize and exist on the surface of mature virions in some cases (e.g., in alphaviruses) [71]. ^b2^ In flaviviruses, the viral fusion protomers form a dimeric complex in the prefusion state [72]. ^c^ In flaviviruses and alphaviruses, proteolytic processing is directed towards another viral protein, which forms a heterodimer with the viral fusion protein while traversing the cellular secretory pathways. In bunyaviruses, proteolytic processing is not well studied [70]. ^d^ Diverse triggers include acidic pH, receptor engagement, or receptor engagement followed by acidic pH. ^e^ Diverse triggers include acidic pH or receptor engagement.

*Class III:* Class III viral fusion occurs in viruses with distant evolutionary origins, such as herpesviruses, rhabdoviruses, and baculoviruses [74]. Unique to class III viral fusion proteins is the reversibility of their conformational transition between the pre-fusion state (at neutral pH) and the post-fusion state (at acidic pH). The fusogenic competency of class III fusion proteins does not depend on any proteolytic event or a viral chaperone protein. With respect to the secondary structural composition of their native fusion subunit, class III proteins appear to resemble a mixture of class I and class II members since they have both α-helical and β-sheet characteristics. In fact, this mix of secondary structural composition is deemed a defining feature of class III viral fusion proteins [74]. Similar to all other viral fusion proteins, the post-fusion conformation of class III members is trimeric.

### 5.4. Viral Fusion Peptides and Their Diversity

Although viral fusion proteins belonging to classes I, II, and III employ distinct mechanisms for membrane fusion, they all feature shared common aspects that include (i) receptor binding, (ii) a priming/triggering event, and (iii) the involvement of fusion-activating domains called fusion peptides (FPs) and fusion loops (FLs). FPs and FLs play a crucial role in initiating membrane fusion. As mentioned previously, following the attachment and the triggering events, the viral fusion proteins undergo conformational changes that expose the fusion peptides/loops for interaction with the target membrane. These relatively hydrophobic domains bind to and insert into the host membranes, establishing a connection between the viral and cellular lipid bilayers. In later stages, class I FPs may interact with the viral fusion proteins’ transmembrane domain (TM) or the membrane-proximal external region (MPER or preTM), potentially stabilizing the hemifusion state and facilitating the formation of fusion pores [75].

FPs generally possess a conserved primary structure within a specific virus family but exhibit significant variations among different viruses within the same class [76]. However, these fusion peptides share several common features, including moderate hydrophobicity, lengths ranging from 15 up to 40 residues, and a notable abundance of alanine and glycine residues [76] (Table 2). FPs are also classified based on their position relative to the proteolytic cleavage site in the fusion subunit. In most class I viral fusion proteins, the FPs are positioned at or near the N-terminus of the fusion unit, whereas those from class II and class III viral fusion proteins tend to be located more internally. However, certain class I viruses such as Ebola virus (EBOV), Avian Sarcoma and Leucosis virus (ASLV), and Marburg virus (MARV) have FPs situated downstream from the N-terminus of the fusion subunit. In coronaviruses such as SARS-CoV-1 and SARS-CoV-2, the subunit harboring the fusogenic region supposedly contains two functional FP segments, one located at the N-terminus and a second FP, referred to as the internal fusion peptide (IFP), positioned more internally, between HR1 and FP [77,78,79,80]. Curiously, SARS-CoV-1 appears to have a third membrane fusion-active sequence known as FP1, found between the Spike protein subunit cleavage [81]. This domain is less conserved across the *Coronaviridae* but well conserved across lineage B betacoronaviruses. FP1 not only exhibits favorable membrane insertion properties (Table 2) but also induces significant alterations in the physical properties of membranes necessary for fusing two lipid bilayers [82,83].

Class I N-terminal fusion peptides are typically α-helical in the pre-fusion state, while class I internal fusion peptides and class II and III fusion peptides often exist as loops connecting two extended β-strands in the pre-fusion state of the fusogen structure. Consequently, fusion peptides from class II and class III fusion proteins are referred to as fusion loops (FLs). N-proximal FPs from class I viral fusion proteins also present a proline residue near or at the center of their sequences, which is a characteristic feature of internal, loop-forming fusion peptides [80], such as the Ebola IFP [84,85]. Proline residues play a crucial role in peptide conformation and fusion activity [84,86]. Furthermore, the presence of proline, glycine, and alanine residues imparts a high level of conformational flexibility to FPs, enabling them to adapt their structures based on their immediate environment. In fact, depending on the peptide-to-lipid molar ratio and lipid composition, the relevant fusion-active secondary structures of class I FPs have been proposed to be predominantly *α*-helix [87,88], *β*-strand [89], disordered [90,91], or even a combination of these conformations [92,93]. Therefore, the lipid-mediated conformational polymorphism of FPs may also play a crucial role in the regulation of membrane fusion.

### 5.5. Viral Fusion Peptides and Their Impact on Biological Membranes

According to established mechanistic models, the interaction between fusion peptides/fusion loops with target membranes goes beyond serving as a mere anchor for cell membranes to viruses. Instead, it induces significant modifications in the physical properties of lipid bilayers, ultimately leading to the merging of virus and cell membranes. These modifications include changes in fluidity, packing, curvature, and hydration [94]. Fusion-activating domains also exhibit the ability to catalyze lipid exchange between adjacent membranes and the mixing of internal contents for fusion pore formation. Additionally, they promote membrane aggregation, permeabilization, and the formation of non-bilayer structures. Various factors, such as lipid composition, ionic strength, specific divalent cations, pH variations, peptide length, and other parameters, can modulate the impact of these peptides on membranes, as well as influence FP structures, oligomeric states, and the depth of FP insertion into lipid bilayers. Biophysical studies involving lipid model membranes and synthetic peptides that mimic the FP and FL regions of entire fusion proteins have contributed significantly to our understanding of the structure and dynamic properties of both the FP/FL and lipid membranes. Notably, a direct correlation has been observed between mutations in these sequences within full-length proteins and the effects of peptide analogs on lipid model membranes [95].

Synthetic peptides corresponding to fusion peptides and fusion loops of viral fusion proteins have an interesting capability to induce membrane fusion in lipid vesicles. However, this effect is observed at higher peptide concentrations in artificial model membranes compared to the corresponding fusion proteins in real cell membranes. As stated above, the fusion process requires close contact between the viral and cellular membranes, lipid exchange between the outer leaflets (hemifusion), and lipid exchange between the inner leaflets, leading to complete fusion and pore formation (Figure 1). The membranotropic effects of fusion peptides/loops that facilitate these steps include lipid aggregation, lipid mixing, membrane permeabilization, and mixing of vesicle internal contents. These effects have been demonstrated for both class I and class II viral fusion peptides/loops [81,82,96,97,98,99,100].

Extensive evidence from numerous studies supports the notion that the insertion of fusion peptides and fusion loops into lipid bilayers induces membrane ordering. The membrane activity of viral fusion peptides/loops can be effectively assessed by measuring the order parameter of lipids using electron spin resonance (ESR) of nitroxide-labeled phospholipids. This approach has been applied to various fusion peptides, including those from influenza [101], HIV-1 [102], SARS-CoV-1 and -2 [81,83,103,104], MERS-CoV [105], EboV [106], and Dengue virus [107]. The ordering of lipid headgroups has been indirectly linked to membrane dehydration [83,108], while the ordering of lipid chains promotes membrane packing, leading to stress in the bilayer that bends it towards the outer monolayer, thereby inducing negative membrane curvature. Differential scanning calorimetry (DSC), ^31^P NMR, and cryo-EM are also commonly used to understand whether the FP induced non-bilayer and isotropic phases, resulting in membrane curvature variation [83,96,109,110,111].

The lipid composition of the membrane plays a crucial role in modulating the secondary structure, membrane insertion depth, location, and fusogenic activity of viral fusion peptides/loops [81,82,83,97,102,112,113,114,115]. It has been observed that cholesterol at concentrations of 30 mol% in the membrane induces a conformational change from α-helical to β-sheet in the HIV fusion peptide [97]. Additionally, lipid phase separation can occur depending on the cholesterol content in the membrane, which may modulate the membrane fusion activity of the fusion peptides [114]. It has also been reported that class II fusion loops exhibit a conserved pocket that recognizes the polar heads of glycerophospholipids (GPLs). The GPL binding facilitates the concentration of cholesterol, thereby enabling the insertion of the fusion loop into the membrane [116].

Another important factor is the presence of divalent cations, which impact the structure and membrane-fusion activity of fusion peptides and loops. In the case of HIV-1 FP, Ca^2+^ does not directly bind to the peptide but induces a conformational change from an α-helix to an antiparallel β-type structure. The absence of Ca^2+^ reduces the infectivity of the virus [117]. Moreover, Ca^2+^ coordination in the FP structure plays an essential role in viral entry [100,103,104,105,106,118,119]. For the EboV IFP, calcium induces inter-vesicle lipid mixing and membrane ordering in lipid model membranes. In the absence of calcium, the fusion activity of this peptide is marginal, but it does promote extensive vesicle permeabilization [100,106]. Similarly, the binding of Ca^2+^ to the FP structures of SARS-CoV-1 and -2, also MERS-CoV Spike proteins induces peptide conformational changes, increased membrane ordering activity, and deeper FP penetration into the membrane [103,104,105,120], yet the presence of Ca^2+^ leads to a decreased lipid mixing activity of these FPs [81].

The reported effects induced via the fusion peptides and fusion loops described above collectively contribute to the following:Reducing the energy barrier for stalk formation by counteracting the repulsive “hydration force” between approaching lipid bilayers;Facilitating the formation of small protrusions in the pre-fusion state, allowing for close intermembrane contact;Promoting lipid mixing and the formation of fusion pores;Providing the necessary free energy for fusion protein refolding, ultimately triggering fusion between viral and cellular membranes.

### 5.6. Viral Fusion Peptides and Their Structural Determinants

The first high-resolution structure of a viral fusion peptide in a model membrane environment was achieved in 2001 for the FP of influenza hemagglutinin (HA) using nuclear magnetic resonance (NMR) [87]. The influenza FP, consisting of a 20-residue-long construct, adopts a distinctive boomerang-like structure in dodecyl-phosphatidylcholine (DPC) micelles, resembling an inverted V-shaped helical amphipathic conformation at the fusogenic pH 5. Conversely, at neutral pH, which corresponds to the non-fusogenic state, the short 3_10_-helix in the C-terminal arm becomes disordered, rendering the peptide inactive. These pH-dependent structural variations demonstrate that the fusogenic conformation of the influenza FP has a deeper insertion into the lipid bilayer core compared to the inactive conformation, implying that the structure and insertion depth play a crucial role in modulating the FP’s function [121,122].

Lorieau et al. investigated the structure of a 23-residue FP construct containing the well-conserved residues Trp^21^-Tyr^22^-Gly^23^. The NMR structure in DPC micelles revealed that these additional residues interact with the N-terminal residues stabilizing a tightly packed, antiparallel helical hairpin with no structural rearrangement between neutral and acidic pH conditions [123]. Lately, an additional G8A mutant resulted in an equilibrium between this closed helical hairpin conformation and two open structures with two stable helices undergoing transitions between L-shaped and extended arrangements [124,125].

Additional structural information regarding class-I FP has been obtained using an entire collection of methods, and X-ray crystallography is an important player in elucidating the viral fusion protein’s structure snapshot. One important example is the crystal structure of the class I Ebola fusion protein trimer in the pre-fusion state, which reveals an antiparallel β-stranded scaffold where the Ebola fusion loop (FL) structure assumes an extended and relatively flat loop-like conformation. This conformation is stabilized via interactions between the FL and neighboring fusion protein residues, as well as a disulfide bond involving conserved cysteine residues C511 and C556 [125]. An isolated 16-residue sequence corresponding to the FL and containing the central hydrophobic tip was studied by Freitas et al. in the membrane-bound state. Solution-state NMR revealed a 3_10_-helix between residues L6 and F12 in sodium dodecyl-sulfate micelles at neutral conditions, while the N- and C-termini remained unstructured [126]. In contrast, Gregory et al. determined the structure of a 54-residue FL construct representing the complete disulfide-bonded internal fusion loop using solution NMR in DPC micelles. The FL structure at non-fusogenic neutral pH displayed an overall elongated and relatively flat shape, similar to the crystal structure of the corresponding sequence in the pre-fusion state of the entire Ebola glycoprotein at pH 8.5 [85]. At fusogenic pH 5.5, a conformational change induced a bend in the loop, resulting in the reorientation of the hydrophobic patch by approximately 90 degrees. This new conformation exhibited helical characteristics within a reversing turn, leading to a more compact structure, which may lead to more fusogenic efficiency.

Most recently, *Coronoviridae* became notably important, as its structural conformation and fusogenic capability comprehension. Solution-state NMR was utilized to obtain the structures of two fusion peptide sequences, FP1 and IFP, from SARS-CoV-1 [127]. FP1 exhibited a V-shaped kinked helical structure in DPC micelles, with a notable bend around a canonical fusion tripeptide F10-G11-G12, a motif highly conserved among FPs of different virus families [128], which likely contributes to its fusion-active structure [129]. On the other hand, IFP adopted a linear helical conformation and a deeper insertion of its more hydrophobic helix into the non-polar region of the micelle compared to FP1 [127]. The authors subsequently reported the NMR structure of a 64-residue fragment of the SARS-CoV-1 Spike protein (Table 2) [130]. The structure is characterized by a short N-terminal helix, a long loop and extended conformations in the central region, and a long helix in the C-terminus. NMR experiments demonstrated the deeper insertion of the central part of FP into the micelle’s hydrophobic core compared to the N- and C-termini residues. The sequence corresponding to the FP domain folds as a helix, whereas that corresponding to FP1 acquires a loop-like conformation and short stretches of helices. This contrasts with their previous findings of an isolated FP1, which exhibited a V-shaped bend helical conformation, indicating potential influence from neighboring residues on FP’s structure and membrane insertion depth. Notably, the FP1 sequences of both SARS-CoV-1 and SARS-CoV-2 Spike proteins appear predominantly disordered in the pre-fusion state, as observed via SPA cryo-EM (PDB IDs: 5XLR and 6VXX).

Analogously, the structure of the complete FP (cFP) sequence of SARS-CoV-2 Spike protein (residues 816–857) in isotropic bicelles was determined by Koppisetti et al. using solution-state NMR [131]. In contrast to FPs found in other class I viral fusion proteins, the cFP exhibited an unusual fold. It consisted of three helices connected by loops, forming a wedge-shaped structure. The first two helices displayed amphipathic properties and were nearly perpendicular to each other, while the C-terminal helix was polar and positioned nearly perpendicular to helix-2. The presence of a 3_10_-helix in the third helix was facilitated via the formation of a disulfide bridge between C840 and C851. Additionally, the first two helices and the connecting loop were deeply inserted into the hydrophobic core of the bicelles, whereas helix 3 appeared to be distanced from the membrane mimetic.

Interestingly, the hydrophobic fusion loops found in class II viral fusion proteins, such as those in the Dengue and Chikungunya virus glycoproteins, adopt virtually similar loop-like conformations between two β-strands in both the pre-and post-fusion states of the full-length E1 fusion proteins [132,133,134]. Solution-state NMR studies revealed that the fusion-active conformation of class II fusion loops in a membrane-bound state closely resembles the pre- and post-fusion states, with minor conformational variations. For instance, the structure of a 15-residue fragment of the Dengue FP in DPC micelles shows that the hydrophobic loop exhibits a similar conformation to the corresponding region in the pre-and post-fusion structures of the full-length glycoprotein [98]. In contrast, the NMR structure of the Chikungunya fusion loop displays a type-I β-turn. In the presence of a membrane mimic, the hydrophobic core of the fusion loop undergoes a conformational rearrangement, leading to increased packing between the aromatic and aliphatic residues, as compared to the corresponding fusion loop region in the crystal structures of the full-length E1 protein [99]. These findings indicate that class II fusion loops undergo spatial reorganization during the membrane fusion process without substantial large-scale structural changes.

### 5.7. Structural Biology and Viral Fusion

While the three classical categories of viral fusion proteins provide a helpful ontological framework for the study of the viral fusion process, they fall short of capturing the observed diversity of the fusion process amongst various viral species within the same class. In addition, this traditional classification does not benefit from the knowledge of the intermediate states between pre-fusion and post-fusion conformations, as such intermediate structures have yet to be captured and resolved structurally. This lack of insight is rooted in the limitations of X-ray crystallography—historically, the most commonly used methodology for determining macromolecule snapshot structures, including viral fusion protein structures—where dynamic protein properties are difficult to capture, particularly for intermediate conformations that are likely to be unstable. Nuclear magnetic resonance (NMR) also falls short in this area, but mostly due to its limitations on protein size and the difficulty of working with proteins embedded in lipid membranes (or membrane mimetics, such as nanodiscs).

Both single particle analysis (SPA) and subtomogram averaging (STA) cryo-EM are powerful techniques in structural biology that have the potential to revolutionize viral fusion protein studies, providing unique insights into their three-dimensional (3D) structure and functional mechanisms [23,135]. With cryo-EM, it is now easier to obtain atomic or near-atomic information on compositionally pure yet conformationally heterogeneous samples. Given the conformational heterogeneity of viral fusion proteins that come with different structural states and conformational changes, cryo-EM is a great tool for capturing and analyzing these distinct conformations and mapping their distribution within viral particles [136,137,138,139]. Indeed, once suitable methods are developed to trap intermediate conformations of viral fusion proteins (embedded in membranes or membrane mimetics) in vitro, SPA methods can be used to resolve distinctly trapped intermediate states from yet-untriggered pre-fusion complexes. Attaining such structures excites not only the many basic scientists who have studied viral fusion processes for the past century but also those clinically oriented scientists interested in viral interactions with host cells, antibody recognition, and immune evasion strategies [136].

By determining the structures of viral fusion proteins bound to their ligands, researchers can elucidate the molecular mechanisms of recognition and binding, providing a foundation for rational drug design and vaccine development [140]. Moreover, virus fusion proteins are prime targets for vaccine development, as they often elicit potent neutralizing antibody responses. Understanding the detailed structure of viral fusion proteins and their antigenic sites via SPA and STA helps in the design of effective immunogens that can induce protective immune responses. By characterizing viral fusion protein structures from different viral strains or variants, researchers can develop broadly neutralizing vaccines that target conserved regions and offer cross-protection against multiple strains. Furthermore, viral fusion proteins are also attractive targets for antiviral drug development. Cryo-EM provides valuable insights into the structural features and functional mechanisms of viral fusion proteins, aiding in the identification and optimization of small molecules or therapeutic antibodies that can disrupt viral entry, fusion, or other essential steps of the viral life cycle.

In summary, cryo-EM is an indispensable tool for studying viral fusion proteins. It allows researchers to visualize their structures, analyze their conformational heterogeneity, study ligand interactions, facilitate vaccine design, and aid in drug discovery efforts, ultimately contributing to our understanding of virus–host interactions and the development of effective antiviral strategies.

## 6. Structurally Dynamic Fusogens—Virally Derived Eukaryotic Fusogens

### 6.1. Cell–Cell Fusion in Placentogenesis

Looking beyond viruses, another notable example of membrane fusion is in the development of placental syncytia in mammalian pregnancy. A major challenge to pregnancy is the delivery of sufficient nutrients to the developing fetus while keeping the maternal immune system tolerant to the foreign body. These challenges are addressed by the syncytiotrophoblast, a multinucleate syncytium found on the outermost layer of the placenta. The syncytiotrophoblast layer is formed via the fusion of the underlying cytotrophoblast cells. Remarkably, placental syncytia are a highly conserved feature across mammalian species, demonstrating the importance of cell–cell fusion in pregnancy [141]. This cell–cell fusion event is catalyzed by fusion proteins derived from retroviruses called syncytins [141,142].

The retroviral life cycle is distinct from most viruses because it requires the integration of viral genetic material into the host genome for productive replication. A consequence of this genomic integration is that retroviral DNA can become permanently fixed within a genome if a germline cell is infected and goes on to be fertilized. As a result, it is believed that approximately 8% of the human genome consists of retroviral DNA [143], with some of these gene products playing important roles in mammalian physiology.

Syncytin proteins are the prototypical example of this phenomenon because the retroviral fusion proteins, and their corresponding cell–cell fusion activity, have been retained throughout millions of years [142,144]. Notably, unique syncytin proteins have been identified in several placental mammals, with some species leveraging multiple syncytin proteins in placentogenesis [145,146,147,148,149,150,151,152,153,154].

Although syncytins are a conserved class of proteins amongst placental mammals, syncytiotrophoblast presentation is diverse across these different species. This has led researchers to hypothesize that structural differences in syncytin proteins are responsible for divergent fusion efficiencies and syncytiotrophoblast diversity. For example, siRNA knockdown of syncytin-2 in trophoblast cells drastically decreases syncytia formation, while syncytin-1 knockdown has a less pronounced effect [155]. This hypothesis is starting to be addressed by talented structural biologists with the determination of the human syncytin structures [156]. Both syncytin structures reveal the canonical trimeric 6HB conformation, which is conserved across retroviral fusion proteins [15,157,158,159] and is a hallmark of class I viral fusion protein post-fusion conformations. The high degree of structural similarity between these fusion proteins, as shown in Figure 4, suggests that structural differences in the 6HB are not responsible for differences in fusion efficiency.

Structural similarity between human syncytins and retroviral fusion proteins, coupled with strong sequence conservation, implies that syncytin proteins from other organisms share a similar 6HB conformation. Structural information is needed to prove this claim. A recent structure-based phylogenetic analysis of class I viral fusion proteins revealed clustering according to the mechanism of membrane fusion [160]. Therefore, a structural analysis of syncytin 6HBs may provide insight into the mechanisms of syncytiotrophoblast formation and how placentogenesis has evolved.

Certain syncytins possess an immunomodulatory activity, which has also been reported for retroviral fusion proteins [161,162]. Immunosuppression is reported for human and mouse syncytins [163] and is thought to promote tolerance of the developing fetus. This immunosuppressive activity is linked to a short 17-amino acid sequence, the immunosuppressive domain (ISD), that has high sequence conservation across syncytins and retroviral fusion proteins [164]. Remarkably, syncytin-2 is immunosuppressive, while syncytin-1 does not demonstrate this activity despite having an 80% sequence similarity to other retroviral ISDs [163]. This trend is also seen in mice, where syncytin-B is immunosuppressive, while syncytin-A is not. Mutagenesis studies using human and mouse syncytins link a single amino acid to ISD activity, demonstrating that Arg at position 14 of the ISD is a hallmark of a non-immunosuppressive fusion protein [163]. These results suggest that immunomodulation via syncytins and retroviral fusion proteins occurs in a sequence-specific manner. Notably, there is a high degree of structural similarity between immunosuppressive fusion proteins and non-immunosuppressive fusion proteins, indicating that the α-helical trimeric leucine-zipper ISD presentation alone is not sufficient for immunomodulatory activity [160]. Several groups have demonstrated that a synthetic peptide homologous to the retroviral ISD, called CKS-17, regulates cytokine expression and activates immune signaling pathways [164,165,166,167]. These studies indicate that CKS-17 requires conjugation to a carrier protein [168] or homo-dimerization via the introduction of a C-terminal cysteine [166] for in vitro activity, suggesting that higher-order structure has an impact on immunomodulatory activity. Further studies are required to uncover the molecular determinants for retroviral and syncytin ISD activity.

### 6.2. Cell–Cell Fusion in Parasite Reproduction

Virally linked fusion proteins can be found in places beyond the placenta, including in the reproductive cycles of human parasites.

Malaria is present in more than 100 countries in tropical and subtropical areas, resulting in ~250 million cases worldwide [169]. The disease in humans is caused by five protozoan species of the genus *Plasmodium*, with *Plasmodium falciparum* being responsible for the most severe cases [169]. Due to this parasite’s resistance to existing drugs and biochemical/molecular similarities between drug targets and their homologs in humans [170,171,172], there is great interest in discovering ways to inhibit transmission between humans and the mosquito vectors that spread the disease. One approach is to target proteins involved in the sexual reproduction portion of *Plasmodium*’s life cycle (Figure 5); the use of small-molecule therapeutics and/or anti-malarial transmission-blocking vaccines at this stage would prevent the proliferation of the parasite within the mosquito vector, staving off further human infections [173].

Studies have revealed that in *Plasmodium*, a gaggle of surface proteins (e.g., P25, P47, P48/45, P230p, and P230) are essential for sexual reproduction [174,175,176,177,178]. However, these proteins are not sufficient for gamete fusion, as an additional male-gamete-specific transmembrane protein is required. Interestingly, this fusion protein is by no means unique to *Plasmodium*. In fact, the first studies of this protein were of its homologs in *Arabidopsis thaliana* and *Lilium longiflorum*, in which they were found to be essential for fertilization [179,180,181]. The two labs working simultaneously on this fertility-associated protein saw fit to give it two names: hapless 2 and generative cell-specific 1 (HAP2/GCS1). HAP2/GCS1 sequences were quickly identified in a variety of lower eukaryotes, including *Plasmodium* and other species of protozoan parasites [181], where they were found to play a role in the fusion between gametes [182]. Today, HAP2/GCS1 is known to be widespread throughout the Tree of Life, although it is conspicuously absent from vertebrates [183,184,185]. It is this potent combination of being essential in parasites yet absent in vertebrates that sets up HAP2/GCS1 as an obvious target for therapeutics [182].

In the interests of sabotaging the reproduction of disease-causing parasites, an understanding of the HAP2/GCS1 function—and the protein structure that supports it—is vital. Sequence prediction pegged HAP2/GCS1 as a single-pass transmembrane protein with N-terminal signal sequences likely to position the protein on the surface of gametes [180,181]; beyond that, a lack of sequence identity to all known fertility factors failed to imply a connection to existing fusion mechanisms. Surprisingly, the use of structural homology modeling showed potential connections to the class II viral fusion proteins [184,186]. This suspicion was confirmed when the structures of HAP2/GCS1 ectodomains from the green algae *Chlamydomonas reinhardtii* [183,184,187] and *A. thaliana* [188], plus an ectodomain fragment from *Trypanosoma cruzi* [188], revealed an obvious structural homology to class II viral proteins from flaviviruses (e.g., Zika virus, dengue fever, West Nile) and alphaviruses (tick-borne encephalitis virus). This structural connection provided the first clues as to the fusion mechanism of HAP2/GCS1.

As mentioned previously, class II viral fusion proteins are dynamic fusogens that transition from pre-fusion dimers into stable post-fusion trimers, inserting hydrophobic fusion loops into an opposing membrane as they do so; given the structural similarities, it would be natural to assume that the fusion mechanism of HAP2/GCS1 relies on this same conformational transition. The first HAP2/GCS1 structures were solved as trimers which, like the post-fusion state of their viral counterparts, cluster the hydrophobic fusion loops together on the same side of the molecule as the proteins’ C-terminal transmembrane helices (Figure 5) [183,184,188]. This provides structural confirmation to biochemical data that HAP2/GCS1 trimerizes following insertion into lipid membranes [184], which has also been seen in vivo during fertilization [189].

The oligomeric state and structure of the pre-fusion conformation remained unclear, with no viral-equivalent dimer of HAP2/GCS1 yet determined. Interestingly, the recombinant protein constructs that yielded the trimeric structures were confirmed to be monomeric in solution prior to crystallization [184,188]. Perhaps this elusive monomer was the pre-fusion state, triggered into trimerizing from the conditions of crystal formation. In support of this hypothesis, a recently solved structure from the extremophile red algae species *Cyanidioschyzon merolae* earned the title of the first monomeric structure of a HAP2/GCS1 ectodomain [190], revealing an extended conformation with all three domains aligned in tandem (Figure 5). However, when it comes to the fusion mechanism, it is unclear if this version of HAP2/GCS1 provides the rule or the exception to it. *C. merolae* lives in low pH environments, and its HAP2/GCS1 requires low pH to trimerize effectively, a reliance not seen in its highly studied homologs [184]. The elucidation of pre-fusion structures from other species is necessary to see how wide reaching this conformation truly is.

At present, the exact triggering event that stimulates HAP2/GCS1 molecules to undergo their conformational change and catalyze fusion remains unclear. As stated previously, acidification is not required to trimerize most of the studied HAP2/GCS1, a stark difference from many viral fusogens. Previous studies using *C. reinhardtii* show that HAP2/GCS1 trimerization requires a species-specific gamete adhesion event, where the adhesin pair FUS1 (plus gamete) and MAR1 (minus gamete) bind to each other, thereby allowing fusion to take place [189]. In a sense, gamete adhesion can be considered a triggering event for HAP2/GCS1, though the molecular mechanism of how this occurs has yet to be elucidated and will likely depend on the species and the gamete adhesion machinery they employ.

Structural biology was able to uncover the link between HAP2/GCS1 and class II viral fusogens, but many questions remain.

*How did viruses and eukaryotes come to share these proteins?* Was the protein transferred to an ancient eukaryotic species during a viral infection, or are the viral proteins eukaryotic in origin? A viral origin similar to syncytins is certainly possible, as retroviral species have been found to carry class II fusogens [191,192]. However, these retroviral class II env sequences are rare and relatively recent compared to ancient HAP2/GCS1. The recent discovery of HAP2/GCS1 sequences in archaeal species further complicates this question [193].

*How conserved are the HAP2/GCS1 mechanisms between different species?* Much of the structural and functional work has been undertaken within a few model organisms, but the widespread nature of HAP2/GCS1 poses questions concerning the translatability of this work between species of such divergent reproductive strategies. For instance, HAP2/GCS1 in *C. reinhardtii* is found only on the plus gamete and is believed to function very similarly to viral fusogens; however, HAP2/GCS1 in *Tetrahymena thermophila* must be expressed on the surface of all seven types of gametes for optimal fusion [194], implying a very different fusion strategy with fusogens on opposing membranes acting in tandem. In *Plasmodium*, two different HAP2/GCS1-like proteins are expressed on the male gamete surface (denoted as PfHAP2 and PfHAP2p) [195]; they are both critical for fertilization and non-redundance in function, adding further confusion to the differing fusion mechanisms between species.

*Can HAP2/GCS1 serve as a transmission-blocking vaccine?* The blocking or depletion of the HAP2/GCS1 fusogen from *Plasmodium* parasites could prevent the spread of malaria between hosts. Indeed, the transmission of the mouse parasite *Plasmodium berghei* was significantly attenuated when mice were injected with a fusion loop-mimicking peptide [171], and antibodies against the C-terminal domain of *Plasmodium* HAP2/GCS1 showed promise at blocking transmission, perhaps by blocking the transition from pre- to post-fusion states (Figure 5) [196]. Whether these strategies can be transferred to humans remains to be seen, and a paucity of whole HAP2/GCS1 ectodomain structures from these species limits much of our structural and functional knowledge to inferences from other organisms.

HAP2/GCS1 sequences have been found in a variety of protozoan parasites beyond just *Plasmodium*. Other Apicomplexa genera also appear to use HAP2/GCS1 for gamete fusion, including *Toxoplasma*, *Babesia*, and *Cryptosporidium* [197,198,199]. In addition, Kinetoplastid genera like *Trypanosoma* and *Leishmania* also contain HAP2/GCS1 sequences [200,201]. Many of these species produce diseases that affect large portions of the world’s population, especially in tropical and subtropical countries. For instance, Leishmaniasis is a neglected tropical disease caused by parasites of the genus *Leishmania*. With around 1.5 million new infections and 20–50 thousand deaths reported annually [202], this vector-borne disease causes deep social and economic consequences, all of which could potentially be mitigated by therapeutics targeting its gamete fusion machinery.

## 7. Mammalian Gamete Fusion: The Golden Goose

### 7.1. What We Know and What We Do Not

The lack of HAP2/GCS1 in vertebrates begs the question: what is the membrane fusion protein behind the sperm–egg fusion process that occurs during vertebrate fertilization? Even though it has been extensively studied, many of the critical steps in this process are not fully understood. As far as is known, the molecular mechanism involves a large number of key proteins, including both egg proteins (JUNO, CD9, CD81, and MAIA) and sperm proteins (IZUMO1, SPACA6, TMEM95, FIMP, DCST1, DCST2, and SOF1) [13,185,203,204,205,206,207,208,209,210,211,212,213,214,215,216,217,218,219,220,221,222,223,224,225,226,227,228,229,230,231,232,233,234,235,236]. A number of these glycoproteins are already structurally characterized, showing conservation with other members of their respective families. However, others are still a mystery and require structural determination to elucidate the fusion mechanism.

Of the solved structures, IZUMO1 and JUNO, and their interaction, have garnered the most attention. IZUMO1 is a type I transmembrane protein localized on the equatorial region of the sperm head. The protein is vaguely shaped like a boomerang, with a four-helix bundle and Ig-like domain positioned on either side of a slightly bent “hinge” region [13,222,227,237]. JUNO is the IZUMO1 receptor expressed on the oocyte’s surface membrane, a globular α/β glycoprotein belonging to the family of folate receptors. The interaction between JUNO and IZUMO1 is known to be crucial for gamete recognition and adhesion [13,222,227,237]. In 2022, Brukman et al. reported that IZUMO1 may also be directly responsible for cell–cell fusion, acting as the sperm fusogen [238]. While this has yet to be validated, the accumulation of IZUMO1-associated oligomerization states [208,210,214,215] in the sperm–egg contact region is critical for the event.

The structure of another sperm-expressed protein vital for gamete fusion, SPACA6, was recently obtained [185,205]. The structural conformation is reminiscent of IZUMO1, making both proteins founding members of a structural superfamily of fertilization-associated proteins. The authors defined members of this “IST superfamily” as having a distorted four-helix bundle and a pair of disulfide-bonded CXXC motifs. Interestingly, another sperm-expressed fusion-associated protein, TMEM95, is also a part of this superfamily. The structure of TMEM95, a type I single-pass transmembrane protein localized in the equatorial segment of sperm, was recently solved. The TMEM95 ectodomain is composed mainly of a distorted four-helix bundle stabilized by similarly placed CXXC motifs [216,239]. The reason for these three proteins of redundant structure but apparently non-redundant function (i.e., IZUMO1, SPACA6, and TMEM95) remains unclear, although TMEM95 appears to be able to interact with oocytes.

The only other solved structures for gamete fusion-associated proteins are CD9 and CD81, both part of the tetraspanin superfamily and expressed on the oocyte surface. Their structures are composed of four transmembrane domains and two extracellular loops, one large (LEL) and one short (SEL) [208,212,216,217,219,235,240]. Structures for FIMP, SOF1, MAIA, and DCST1/2 remain elusive.

### 7.2. A Potential Mechanism

The mechanism for sperm–egg attachment and fusion remains unclear, but several studies have indicated that the above-mentioned proteins interact to form a fusogen complex (Figure 6). The initial step (sperm–egg attachment) occurs via the binding of IZUMO1, expressed in the sperm surface, and its receptor JUNO, which is localized in the oocyte membrane. It is known that monomeric JUNO induces conformational domain changes in IZUMO1 and that IZUMO1 forms a multimer in the adhesion area. The trigger for this multimerization is under investigation, with protein disulfide-isomerases (PDIs) observed on the sperm surface as a potential lead [203].

The IZUMO1–JUNO high-affinity binding gives rise to an accumulation of tetraspanin CD9 and CD81 at the interface between the sperm and egg, which enables closer physical contact between them. At this point, JUNO is shed from the membrane, and likely, other fusion-associated proteins (such as SPACA6, TMEM95, FIMP, and SOF1) are recruited for the fusion pore formation [203]. However, which of these players are involved in this stage and what they do remains to be seen. Indeed, the entirety of the fusion pore formation mechanism is not fully understood. It has been hypothesized that a fusogen, perhaps IZUMO1 or a combination of factors working together, catalyzes the hemifusion intermediate formation, where the sperm and egg’s outer leaflet membrane bilayers mix, first followed by the inner bilayer leaflets.

### 7.3. What Can Cryo-EM Do for Gamete Fusion Research?

Recently, with the new advances in sample preparation, imaging, and data processing, cryo-EM has been helping to elucidate the atomic details of the fertilization process. Cryo-EM has provided significant insight into gamete ultrastructure, but the harshness of past sample preparation methods has precluded observing at a truly molecular level [241,242,243,244,245,246,247]. However, SPA, such as demonstrated by Umeda et al. for the CD9-EWI-2 complex [240], and cryo-electron tomography (cryo-ET) for the uromodulin (UMOD)/Tamm–Horsfall complex (a common sperm-binding region at the interface between subunits) [248], has shown how analysis of high-pressure frozen specimens is helping to understand the molecular mechanisms on the surface of gametes [242,243,244].

Studies using cryo-EM to visualize complexes associated with cell and organelle mobility are being routinely used [243,244,249], and the recent advancements in cryo-ET provide an opportunity to image cells in a near-native state and at unprecedented levels of detail. New and emerging cellular EM techniques are poised to rekindle the exploration of fundamental questions in mammalian reproduction, especially phenomena that involve complex membrane remodeling and protein reorganization. As an example of what can be achieved, advances in STA have revealed mammalian sperm-specific protein complexes within the microtubules, the radial spokes, and nexin–dynein regulatory complexes. The locations and structures of these complexes suggest potential roles in enhancing the mechanical strength of mammalian sperm axonemes and regulating dynein-based axonemal bending. These findings are associated with possible regulation of the sliding of each microtubule doublet and may underlie the asymmetric beating of mammalian sperm [250].

## 8. Conclusions and Perspectives

Structural biology has played a crucial role in understanding the mechanisms of membrane fusion, a fundamental process in various biological events such as intracellular trafficking, cell–cell communication, neurotransmission, muscle tissue formation/repair, viral entry, and sexual reproduction.

A combination of X-ray crystallography, cryo-EM, and NMR spectroscopy has allowed for the determination of high-resolution 3D structures of the proteins and protein complexes involved in membrane fusion. These structures provide insights into the molecular architectures of fusion proteins, their domain organizations, and the arrangement of functional motifs critical for fusion. As membrane fusion involves conformational changes in fusion proteins that facilitate the merger of lipid bilayers, structural biology techniques enable the characterization of these conformational changes by comparing the structures of fusion proteins in their pre-fusion and post-fusion states. Understanding the structural transitions provides mechanistic insights into how fusion proteins undergo dramatic conformational rearrangements to drive membrane fusion. With X-ray crystallography as the former main technique for studying this dynamic process, the static snapshots likely failed to capture the myriad of potential intermediate states present throughout membrane fusion. Cryo-EM, especially SPA and STA, can capture and determine the structures of these fusion intermediates. By visualizing the transient states during fusion, researchers can dissect the sequence of events and propose mechanistic models that explain the fusion process at atomic resolution.

A full understanding of these processes is vital for therapeutic interventions in various diseases, including viral infections, neurological disorders, and malaria. Structural biology provides a foundation for rational drug design by revealing the atomic details of fusion protein structures and their interactions with small molecules or antibodies. This information enables the identification of potential drug-binding sites and aids in the development of inhibitors that can block membrane fusion and its unwanted results.

## Figures and Tables

**Figure 1 biomolecules-13-01130-f001:**
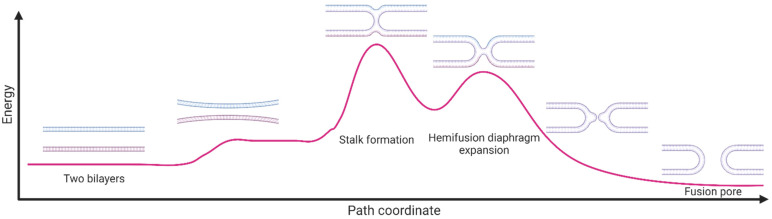
Membrane fusion mechanism. The process of membrane fusion, which is a crucial cellular mechanism involved in various biological processes, is divided into four distinct stages, each representing a key step in the fusion process. At the initial stage, two separate lipid bilayers are shown in close proximity to each other. The membranes are depicted as curved structures with different colors, highlighting their distinct identities. For membrane docking and recognition, the two opposing membranes undergo a series of molecular interactions to facilitate fusion. Proteins and other molecules aid in the recognition of specific fusion partners and ensure proper alignment of the membranes. The final stage showcases the actual fusion event between the membranes. The lipid bilayers seamlessly merge, resulting in the formation of a single continuous membrane structure. This fusion process enables the mixing of cellular contents and facilitates the exchange of molecules between the previously separate compartments. Created with BioRender.

**Figure 2 biomolecules-13-01130-f002:**
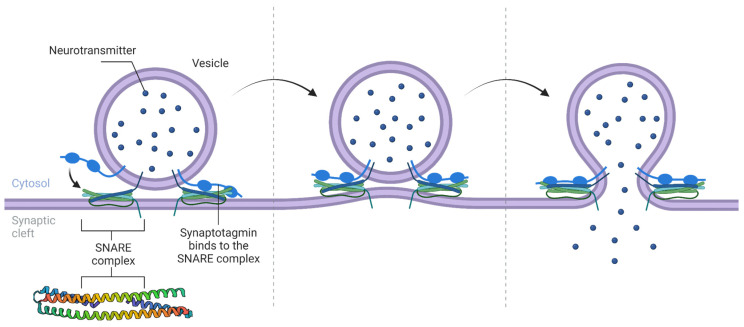
Mechanisms of SNARE proteins in membrane fusion. The process begins with the docking of a vesicle to the target membrane. The vesicle and target membranes come into close proximity, facilitated via specific interactions between SNARE proteins (PDB ID 2C5J) present on both membranes. SNARE proteins on the vesicle membrane (v-SNAREs) and the target membrane (t-SNAREs) interact with each other, forming a highly stable and tightly coiled complex known as the SNARE complex. This complex consists of a four-helix bundle, with the v-SNARE and t-SNAREs contributing two helices each. The SNARE complex undergoes a process called “zippering”. This involves the progressive and tight association of the four helices, pulling the vesicle and target membranes closer together. The energy released during this zippering process helps overcome the repulsive forces between the membranes, leading to their fusion. As the SNARE complex completes its zippering, the lipid bilayers of the vesicle and target membranes merge, allowing the contents of the vesicle to be released into the target compartment. This process is accompanied by the mixing of transmembrane proteins and lipids, facilitating the exchange of molecules between the two compartments. After membrane fusion, the SNARE complex disassembles with the help of ATPase NSF (N-ethylmaleimide-sensitive factor) and its cofactor SNAPs (soluble NSF attachment proteins). This disassembly step prepares the SNARE proteins for subsequent rounds of membrane fusion. Created with BioRender.

**Figure 3 biomolecules-13-01130-f003:**
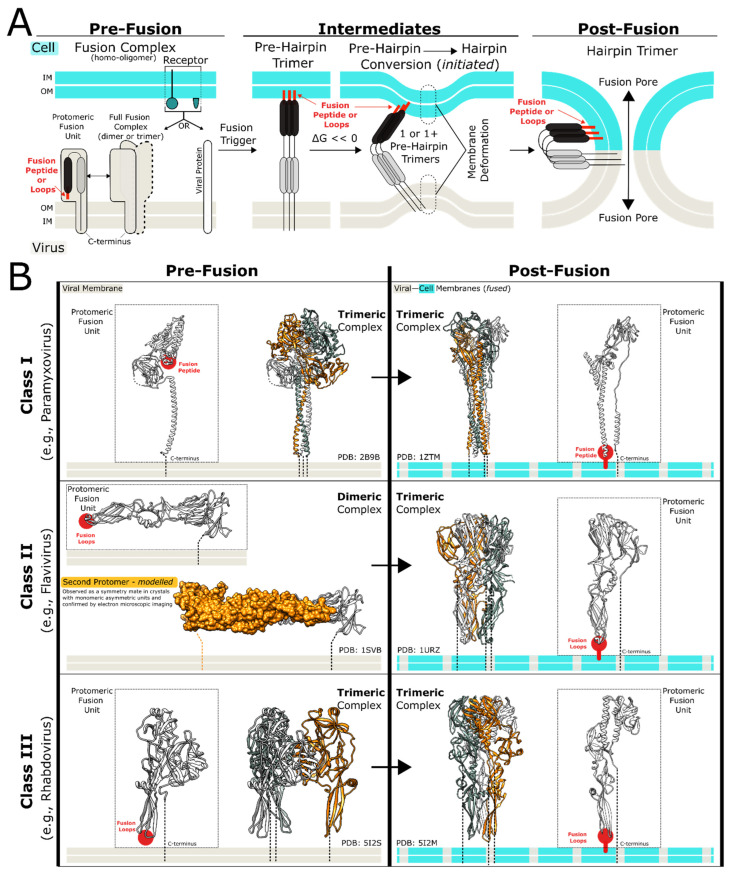
Viral—cell membrane fusion mechanisms. (**A**) A generalized schematic of the viral fusion to cellular membranes. Homo-dimeric or homo-trimeric (and in some cases heterodimeric) pre-fusion complexes initiate attachment via cellular receptors, such as cellular membrane proteins or surface sugar moieties. Following a fusion trigger event (full receptor engagement, acidic pH, or both), the pre-fusion complex undergoes conformational changes that result in the insertion of the hydrophobic fusion peptide or a series of fusion loops into the cellular membrane, forming a high-energy pre-hairpin trimer intermediate. The pre-hairpin complex undergoes stabilization via the collapsing of its N- and C-termini, hence pulling the two membranes towards one another, resulting in the mixing of viral and cellular membranes and the formation of the fusion pore. (**B**) Representative structures of class I, II, and III viral fusion complexes in pre-fusion and post-fusion states. The pre-fusion complexes of class I and III are trimeric, while class II complexes are either homo-dimeric (shown here) or hetero-dimeric in nature and run parallel to the viral membrane (heterodimers can, in some cases, trimerize in the mature form). All classes possess trimeric post-fusion conformations. For a side-by-side comparison of the fusion complexes of the three classes, see Table 1.

**Figure 4 biomolecules-13-01130-f004:**
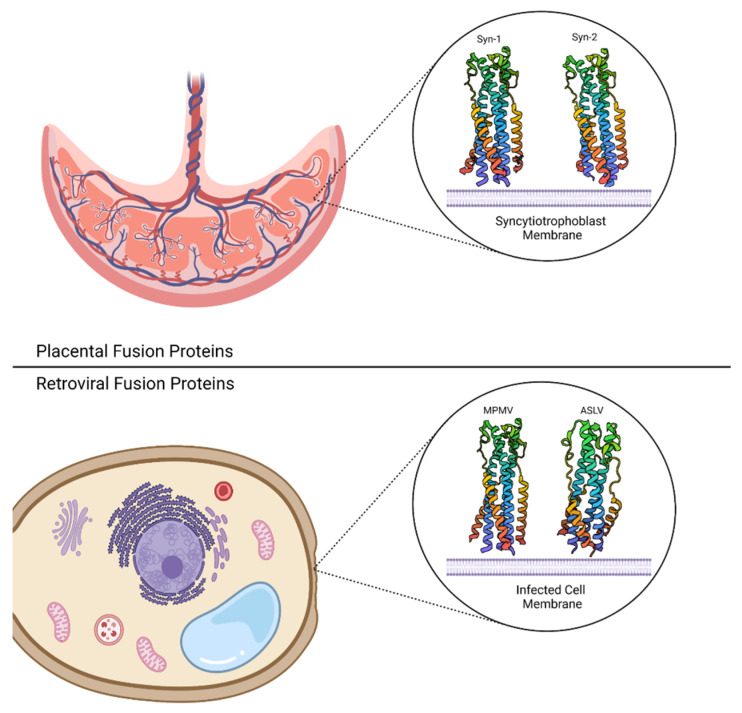
Placental and retroviral fusion protein six-helix bundles (6HBs). The Syncytin-1 (PDB ID: 6RX1) and Syncytin-2 (PDB ID: 6RX3) 6HBs are oriented relative to the placental syncytiotrophoblast membrane (top), while the Mason-Pfizer Monkey Virus (PDB ID: 4JF3) and Avian Rous Sarcoma Virus (PDB ID: 4JPR) 6HBs are shown relative to the infected cell membrane (bottom). Both placental and retroviral proteins form the canonical post-fusion 6HB conformation following receptor recognition and/or pH acidification. Created with Biorender.

**Figure 5 biomolecules-13-01130-f005:**
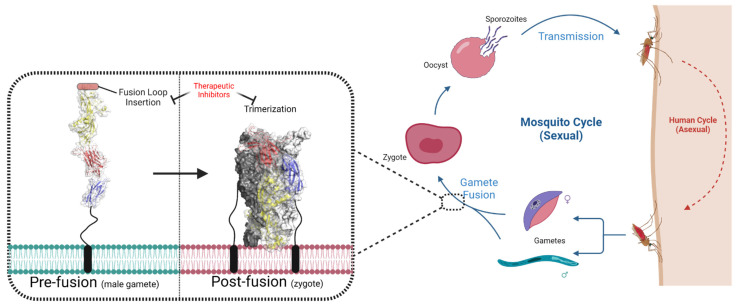
Membrane fusion as part of the *Plasmodium* life cycle. The sexual reproductive portion of the *Plasmodium* life cycle takes place in the mosquito vector, where haploid gametes merge to form diploid zygotes. This process requires the fusion protein HAP2/GCS1 to undergo a conformational change from a pre-fusion state (likely monomeric) to a post-fusion trimer; in the process, the relative orientations of domains 1 (red), 2 (yellow), and 3 (blue) also change. Since no solved structures of *Plasmodium* HAP2/GCS1 exist, representatives of the pre-fusion and post-fusion state are taken from *Cyanidioschyzon merolae* (PDB: 7S0K) and *Chlamydomonas reinhardtii* (PDB: 6E18), respectively. Therapeutic inhibitors targeting either the insertion of hydrophobic fusion loops into the membrane or the trimerization event could potentially prevent the formation of zygotes, thereby preventing the formation of haploid sporozoites that instigate further infections. Created with BioRender.

**Figure 6 biomolecules-13-01130-f006:**
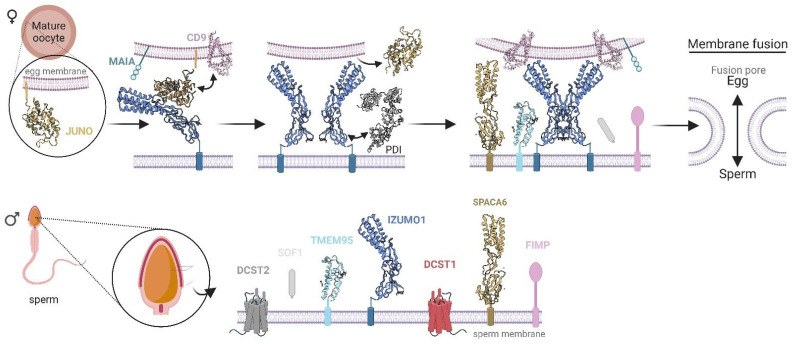
Key proteins involved in sperm-egg fusion. JUNO (purple) and MAIA (light green) are localized in the egg’s membrane in females. In males, IZUMO1 (blue), SPACA6 (yellow), TMEM95 (light blue), and FIMP (pink) are in the equatorial regions of the sperm. DCST1 (red) and DCST2 (dark grey) are transmembrane proteins, and SOF1 (light grey) is a secreted protein. During the attachment phase, IZUMO1 recognizes its receptor JUNO and their binding gives rise to an accumulation of CD9, which enables closer contact between the egg and the sperm. PDI has been observed on the sperm surface in colocalization analysis. After the binding, a conformational change in IZUMO1 occurs, which becomes oligomeric and is thought to recruit proteins to the gamete fusion complex (e.g., SPACA6, TMEM95, SOF1, unidentified fusogen); however, each protein’s roles are still unclear. The fusion pore formation mechanism is not fully understood, but it is believed that a fusogen catalyzes the hemifusion intermediate formation, where the sperm and egg’s outer leaflets membrane bilayers mix, first followed by the inner bilayer leaflets mix, originating the fusion pore. Created with BioRender and adapted from [203].

**Table 2 biomolecules-13-01130-t002:** Primary structure and free energy change of peptide transfer from the water phase to the lipid bilayer surface for fusion peptides/loops of different classes of viral fusion proteins.

Class	Virus ^a^	Name ^b^	Sequence	# Residues	Δ*G_IF_* (kcal/mol) ^c^
I	IFV-A	FP	GLFGAIAGFIENGWEGMIDGWYG	23	−2.52
	PIV5	FP	FAGVVIGLAALGVATAAQVTAAVALV	26	0.04
	HIV-1	FP	AVGIGALFLGFLGAAGSTMGARS	23	−2.29
	ASLV	IFP	GPTARIFASILAPGVAAAQALREIERLA	28	5.94
	SARS-CoV-1	FP1	MYKTPTLKYFGGFNFSQIL	19	−3.07
		FP	SFIEDLLFNKVTLADAGF	18	1.21
		cFP	SFIEDLLFNKVTLADAGFMKQYGECLGDINARDLICAQKF	40	5.89
		IFP	MIAAYTAALVSGTATAGWTFGAGAALQIPFAMQMAYRF	38	−4.46
	SARS-CoV-2	FP	SFIEDLLFNKVTLADAGF	18	1.21
		cFP	SFIEDLLFNKVTLADAGFIKQYGDCLGDIAARDLICAQKF	40	4.77
		IFP	MIAQYTSALLAGTITSGWTFGAGAALQIPFAMQMAYRF	38	−5.20
	EBOV	IFP	GAAIGLAWIPYFGPAAE	17	−1.3
	MARV	IFP	LAAGLSWIPFFGPGI	15	−4.45
	NDV	FP	FIGAIIGSVALGVATAAQITAA	22	−0.45
II	DENV-1	FL	DRGWGNGCGLFGKGSL	16	−0.70
	SFV	FL	VYTGVYPFMWGGAYCFCDS	19	−3.86
	CHIKV	FL	VYPFMWGGAYCFCDTENT	18	−2.11
III	HSV	FL	VWFGHRY/RVEAFHRY	15	0.70
	AcMNPV	FL	YAYNGGSLDPNTRV/VKRQNNNHFAHHTCNK	30	8.36

# Specific number of residues present in the sequence. ^a^ IFV-A: Influenza A Virus; PIV5: Parainfluenza Virus 5; HIV-1: Human Immunodeficiency Virus Type 1; ASLV: Avian Sarcoma and Leucosis Virus; SARS-CoV: Severe Acute Respiratory Syndrome Coronavirus; EBOV: Ebola Virus; MARV: Marburg Virus; NDV: Newcastle Disease Virus; DENV-1: Dengue Virus type 1, SFV: Semliki Forest Virus, CHIKV: Chikungunya Virus; HSV: HerpesSimplex Virus and AcMNPV: Autographa californica Multiple Nucleopolyhedrovirus. ^b^ FP: fusion peptide; cFP: complete fusion peptide; IFP: internal fusion peptide; and FL: fusion loop. ^c^ The interfacial hydrophobicity score (free energy change from water to the lipid bilayer interface) was determined according to the Wimley–White whole-residue scale.

## Data Availability

No new data were created or analyzed in this review. Data sharing is not applicable to this article.

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
