# Peer review of "A Frame-by-Frame Glance at Membrane Fusion Mechanisms: From Viral Infections to Fertilization"

_biomolecules, 2023, doi:10.3390/biom13071130_

Round 1

Reviewer 1 Report

Azimi et al. described the current situation on the analysis of the mechanism of membrane fusion in this review. They covered the broad areas of biological events involving membrane fusion processes such as intracellular trafficking of vesicles, viral and parasite infection, and fertilization. They also described the contribution of structural analyses of involved proteins for our understanding of the membrane fusion mechanism. The readers will obtain the recent landscape of the field through this review. Though the readers may not get the “frame-by-frame” depiction of the fusion mechanism, certainly this review will help readers in various levels. The manuscript is well organized and easy to read.

A few points should be addressed to further improve the readability of the review.

  1. In Table 1 the spacing of the word is not appropriate in column, and it hinders the readability. (in Class I, the space between Homo-trimeric and (or …) in the second line is too broad.
  2. In Figure 6 legend; “SOF1 is a secret protein” should be “SOF1 is a secreted protein”.

As described above.

Author Response

Point-by-point response to reviewers:

We would like to cordially thank both reviewers for their time and effort in the matter of our manuscript and for their positive and encouraging comments. Their constructive feedback helped us improve the quality of our work through the CORRECTIONS detailed below. Note that the revised sections, as per instructions from the editor, are highlighted in the text (in yellow in the revised version - please see the attached file) for ease of identification.

Correction #1 (requested by Reviewer 1)

***Correction for Comment regarding the title of the manuscript.

Response: We thank Reviewer #1 for the highlight about  “Though the readers may not get the “frame-by-frame” depiction of the fusion mechanism, certainly this review will help readers on various levels.”. Our main ideas about this title were to highlight the importance of understanding the membrane fusion stepwise and also the usage of structural methods (such as X-ray crystallography and cryogenic electron microscope – which uses framing data acquisition) to fulfill this important event. Therefore, we believe the title may be adequate for the purpose of this review.

 Correction #2 (requested by Reviewer 1)

***Correction for Table 1; text spacing.

Response: The table was properly redesigned and it should be appropriately fixed and highlighted in yellow.

Correction #3 (requested by Reviewer 1)

***Correction for the misspelling of the word “secreted”

Response: We thank Reviewer #1 for finding this typo, which was properly fixed and highlighted in yellow.

Reviewer 2 Report

This is an extensive and good review of viral fusion mechanisms. It includes intracellular cellular fusogens that regulate vesicle transit (SNAREs), viral fusogens (classes I, II and III) and eukaryotic fusogens that fuse cells together (syncytins, HAP2 and the sperm-egg fusion system).I particularly appreciated the sections devoted to cell fusion but I found the sections on viral fusion could be improved. Particularly, I found the sections devoted to fusion peptides/loops (lines 414-630) too long and unclear. I think the reader would appreciate a synthesis of these sections.

The comments that follow refer to these sections and in particular to the class II fusion proteins, where I have found a number of inaccuracies that deserve to be corrected: 

1) In lines 371-400 there are a number of inaccuracies. Class II fusion proteins have been identified in flaviviruses (only one genus), alphaviruses and several families of bunyaviruses, not just phleboviruses (see PMID: 28433053). In the prefusion form they form a heterodimer with a accompanying protein. Only flaviviruses are homodimers and these are the exception, not the rule. The same can be said of their maturation and activation, for example furin maturation occurs in flaviviruses and alphaviruses but not in most bunyaviruses. The authors should correct the prefusion oligmeric state and proteolytic processing parts for class-II FPs in table 1. 

2) Lines 415-417. Although the authors mentioned this later, fusion peptides (FP) are only in class I and not in classes II and III, where there are fusion loops. at this point in the review is unclear and the following paragraphs refer exclusively to class I protein fusion peptides. Please, correct this also in Fig. 1

3) Lines 610-630. Here I have several comments. In the class II fusion proteins of alphavirus, rubella virus and several bunyaviruses (see PMIDs: 21124458, 23292515, 32937107, 27783711, 34077751) the conformation of the fusion loop in the prefusion conformation (in complex with the accompanying protein) is different from the conformation of the fusion loop in the postfusion conformation (in the absence of the accompanying protein and at acidic pH) and this (probably) is the conformation that the fusion loop acquires when inserted into membranes. In addition, and this is an important difference with class I FPs, class II fusion loops not only insert hydrophobic residues into the aliphatic section of cell membranes, but they have a conserved pocket that recognises the polar heads of glycerophospholipids (PMID: 29097548)

I hope these comments will help the authors to improve their review. 

Author Response

Point-by-point response to reviewers:

We would like to cordially thank both reviewers for their time and effort in the matter of our manuscript and for their positive and encouraging comments. Their constructive feedback helped us improve the quality of our work through the CORRECTIONS detailed below. Note that the revised sections, as per instructions from the editor, are highlighted in the text (in yellow in the revised version - please see the attached file) for ease of identification.

Correction #4 (requested by Reviewer 2)

***Correction for several inaccuracies within Lines 371 to 400 (= section on Class II viral proteins)

Response: We are very thankful to Reviewer 2 for the several clarifying comments regarding the section on Class II viral fusion mechanisms (lines 371-400) and requested changes to the text and Table 1. The corrections are highlighted in the revised text:

We indicated that bunyaviruses (and not only the phlebovirus genus of bunyaviruses) possess a Class II viral fusion mechanism. We included the reference article provided by the Reviewer #2.

We highlighted that the prefusion state of the Class II viral fusion proteins is only homo-dimeric among flaviviruses, while other Class II viruses (e.g., alphaviruses) have hetero-oligomeric states. We included appropriate references in this regard. We also made explicit clarifying changes to the caption of Figure 1 and Table 1.

We made corrections to Table 1, row “Proteolytic Processing of Fusion Proteins”, to address the comment regarding the furin maturation pathways of Class II viral fusion proteins.

Correction #5 (requested by Reviewer 2)

***Correction for the chronology of presenting the notion of fusion peptide vs fusion loops

Response: We thank the Reviewer for bringing attention to this important detail. Indeed, fusion loops are fusion peptides that assume a loop-like conformation. It is worth noting that fusion loops are not limited to class II and class III viral fusion proteins. Internal fusion peptides from class I viral fusion proteins, such as the one found in the Ebola virus glycoprotein, also exhibit a loop-like conformation and are considered fusion loops [see https://doi.org/10.1073/pnas.1104760108]. To avoid further misunderstanding, the sentences have been revised as recommended.

Correction #6 and onwards (requested by Reviewer 2)

***Correction for a synthesis of ideas described in Lines 414 to 630 to make it shorter and to the point

Response: The referee’s observation is correct. However, as we mentioned in the final sentence of this paragraph, we intended to convey that class II fusion loops, such as those found in the Dengue virus and Chikungunya virus, do not undergo significant structural changes between the crystallographic structures of the prefusion and postfusion states, and the NMR structure of the micelle-bound state. As stated by Melo et al., when comparing the structure of the Dengue FL in micelles with its crystallographic structures, “The r.m.s.d. between the NMR and crystallographic structure is 3.41 Å; the r.m.s.d. between the pre-and post-fusion states of this segment in the crystal structures is 0.24 Å” (PMID 19619560). In addition to the results discussed in our manuscript regarding the fusion loops of the Dengue and Chikungunya viruses, it is worth noting that the hydrophobic tip of the class I fusion loop from the Ebola virus bends nearly 90 degrees in the membrane-bound state compared to the prefusion state, while maintaining the overall loop between two β-strands conformation [please refer to https://doi.org/10.1073/pnas.1104760108].

We also appreciate the referee’s suggestion – “In addition, and this is an important difference with class I FPs, class II fusion loops not only insert hydrophobic residues into the aliphatic section of cell membranes, but they have a conserved pocket that recognises the polar heads of glycerophospholipids (PMID: 29097548).” and we added a sentence clarifying this topic including the reference.
